# The Puf-A Protein Is Required for Primordial Germ Cell Development

**DOI:** 10.3390/cells11091476

**Published:** 2022-04-27

**Authors:** Chi-Fong Ko, Yi-Chieh Chang, Huan-Chieh Cho, John Yu

**Affiliations:** Institute of Stem Cell and Translational Cancer Research, Chang Gung Memorial Hospital, Taoyuan 333, Taiwan; kochifong@ntu.edu.tw (C.-F.K.); d9901101@gmail.com (Y.-C.C.); cchaser2002@gamil.com (H.-C.C.)

**Keywords:** Puf-A, nucleophosmin (NPM1), p53, primordial germ cells (PGCs), proliferation, apoptosis, PGC maintenance

## Abstract

Puf-A, a nucleolar Puf domain protein, is required for ribosome biogenesis. A study of Puf-A in zebrafish has shown that Puf-A is highly expressed in primordial germ cells (PGCs) and participates in PGC development. However, it remains unclear how Puf-A governs PGC development in mammals. Here, we generated transgenic mice carrying inducible *Puf-A* shRNA and obtained double heterozygous mice with *Puf-A shRNA* and *Oct4-EGFP* to examine the behavior of PGCs. It was found that the knockdown of *Puf-A* led to the loss of a considerable number of PGCs and a slowdown of the movement of the remaining PGCs. Puf-A and NPM1 colocalized in clusters in the nuclei of the PGCs. The silencing of *Puf-A* resulted in the translocation of NPM1 from nucleolus to nucleoplasm and the hyperactivation of p53 in the PGCs. The PGCs in *Puf-A* knockdown embryos showed a significant increase in subpopulations of PGCs at G1 arrest and apoptosis. Moreover, the expression of essential genes associated with PGC maintenance was decreased in the *Puf-A* knockdown PGCs. Our study showed that Puf-A governed PGC development by regulating the growth, survival, and maintenance of PGCs. We also observed the alterations of NPM1 and p53 upon *Puf-A* knockdown to be consistent with the previous study in cancer cells, which might explain the molecular mechanism for the role of Puf-A in PGC development.

## 1. Introduction

Puf-A, a member of the Pumilio RNA-binding protein family (PUF family), is universally expressed in eukaryotes, including yeast, plants, and humans. These proteins have an RNA-binding domain, which is composed of Pumilio (PUM) repeats and is involved in post-transcriptional control by binding to the cis-regulatory elements of gene transcripts [1].

These PUF family proteins can be classified into three groups by their molecular function and structure arrangement:I.the classical PUF proteins which control mRNA translation by binding to 3′UTR,II.the Nop9 orthologues for small subunit pre-rRNA processing, andIII.the PUM3 orthologues for the biogenesis of the large ribosomal subunit [2].

In mammals, there are two classical PUF proteins, PUM1 and PUM2 [3]. Many studies reveal that PUM1 and PUM2 are translational repressors and play crucial roles in multiple biologic events, including embryonic development, fertility, neurological defect, etc. [4,5,6,7,8,9,10]. NOP9 promotes the cleavage of small subunit pre-rRNAs and the maturation of 18S rRNA in yeast [11,12]. PUM3 proteins are predominantly found in the nucleolus for large ribosome biogenesis. The nucleolar Puf6, the PUM3 ortholog, in yeast coordinates with Loc1 and Rpl43 for the biogenesis of the large ribosomal subunit [13,14]. In budding yeast, Puf6 also functions as a translation repressor of *ASH1* via binding directly to its mRNA 3′UTR during anaphase of the cell cycle [15]. Puf-A, the PUM3 orthologous protein, is conserved in zebrafish, mice, and humans [16]. Computer modeling of the crystal structures of Puf-A demonstrates the interaction between Puf-A and 5.8S rRNA [17]. Based on immunofluorescence staining and immunoprecipitation studies, Puf-A was also shown to interact with another nucleolar protein, nucleophosmin (NPM1), to promote ribosome biogenesis in cancer cells [18]. Although human Puf-A and yeast Puf6 are nucleolar proteins and participate in ribosome biogenesis, they have only 24% homolog, implying that their biological functions might be different.

The previous study describes how the loss of *Puf-A* in zebrafish embryos leads to a reduction in primordial germ cell (PGC) numbers and abnormal migration [16]. These results prompted us to study the role of Puf-A in mouse germ cells during gonad development. PGCs are the founder cells of germline lineage essential for reproduction, and they differentiate to form the gonads. In mouse embryos, the PGCs originate from proximal epiblast cells, which receive BMP4 signaling from extraembryonic ectoderm for determining their fate between embryonic days E6.25 and E6.5 [19]. Early PGCs express the germ cell lineage marker Blimp1 until E7.0 [20,21]. After induction by Blimp1 to repress the somatic program and to express pluripotency genes, these PGC precursors increase in number from E7.25. The PGCs are first detectable at E7.5 in a small group of cells and then move to the posterior of the primitive streak in the extraembryonic mesoderm around the base of the allantois at E8.5 [22]. From the allantois, the PGCs become motile and start on a migration route through the hindgut endoderm (E9.0), along the dorsal mesentery (E9.5), finally reaching the genital ridges at E10.5-E11.5. Throughout their migration phase, the PGCs continue to exhibit extensive mitosis after settling in the genital ridges. Afterwards, the PGCs colonize the gonads and the number of PGCs is greater than 25,000 cells at E13.5 [23].

In the present study, we generated transgenic mice carrying the inducible *Puf-A* shRNA. The PGCs from the *Puf-A* knockdown mouse embryos became less motile starting from E9.5, thus causing the decrease in cell proliferation during E10.5 and 13.5 and the suppression of PGC migration. In addition, the PGCs with decreased cell motility exhibited apoptotic cell death and a lack of mitotic division. Puf-A was found to co-localizedwith NPM1 in the nucleolus of the PGCs in E10.5 embryos. The loss of *Puf-A* resulted in nucleoplasmic translocation of NPM1 and aberrant activation of p53, presumably accounting for the impaired cell-cycle progression and enhancement of apoptosis in knockdown mouse embryos. The knockdown of *Puf-A* also blocked PGC migration and loss of germ cell identity. Therefore, these results suggest that Puf-A is required for gonad development and participates in the growth, survival, and PGC maintenance during gonad development.

## 2. Materials and Methods

### 2.1. Construction of Plasmids

To construct a *Puf-A* inducible knockdown plasmid, a pair of complementary oligonucleotides of *Puf-A* were synthesized and annealed to generate a 71 bp long double-stranded DNA (Figure 1A), which was subcloned into the XhoI and HindIII sites of the vector, downstream of the modified *P*_TREmod/U6_, to yield the 7.7 kb pPGK-tTs-TRE-Puf-A shRNA plasmid for transgenic construction (Figure 1A). This vector plasmid contained:I.the promoter of phosphoglycerate kinase promoter (*P*_PGK_)II.*P*_PGK_ for the expression of tetracycline-controlled transcriptional suppressor (tTS)III.tTS, which corresponded to the sequence for Tet repressor protein (TetR) fused with the KRAB-AB silencing domain of the Kid-1 protein (SD^Kid−1^), representing a powerful transcriptional suppressor for shRNA expression [24,25] through its binding to the tetO sequencesIV.*β*-globin polyAV.the modified Tet-responsive Pol III hybrid promoter (*P*_TREmod/U6_) with tetO sequences (Figure 1A).

### 2.2. Transgenic Mice Carrying the Inducible Puf-A Knockdown Construct

To generate *Puf-A* inducible knockdown transgenic mice, the 7.7 kb pPGK-tTs-TRE-*Puf-A* shRNA plasmid was excised with EcoRV and HindIII to yield a 3.4 kb *Puf-A* shRNA transgene fragment. Transgenic mice were generated by the microinjection of this 3.4 kb *Puf-A* shRNA fragment into the pronucleus of fertilized eggs from superovulated C57BL/6 mice (Level Biotechnology Inc., Taipei, Taiwan). The embryos of *Puf-A* shRNA^+/–^
*Oct4-EGFP*^+/–^ administrated with doxycycline were referred to as the “*Puf-A* knockdown,” and the embryos of *Puf-A* shRNA^+/–^ *Oct4-EGFP*^+/–^ without doxycycline treatment and *Oct4-EGFP*^+/+^ treated with doxycycline were referred to as “shRNA control” and “Dox control,” respectively.

### 2.3. Generation and Genotyping of Transgenic Mice

Fragments were cut from the tails of mice and collected in a 1.5 mL Eppendorf tube containing 300 μL of lysis buffer (125 μg/mL proteinase K, 100 mM Tris-Cl (pH 8.3), 100 mM KCl, 0.02% gelatin, and 0.45% Tween 20). Afterward, these tail lysates were incubated at 56 °C for 1 h, boiled for 10 to 15 min, and centrifuged at 12,000 rpm for 5 min at 4 °C. The supernatant (150 μL) was collected and stored at −20 °C until used as the DNA solution. Then the mice were genotyped by PCR analysis of the tail genomic DNA with primer #1 (5′-GGC TGT GCT CTT TAC TCG GG-3′) and primer #2 (5′-CCA AAC CGG GCC CCT CTG CT-3′) to amplify a 730 bp fragment of the transgene. The non-transgenic littermates served as controls. The PCR reaction mixture (25 μL) was contained 2 μL of DNA solution with 1X PCR buffer, 1.5 mM MgCl_2_, 300 pmol dNTPs, 12 pmol of primers, and 0.25 U Taq DNA polymerase (Takara Bio. Inc., Shiga, Japan). The PCR was performed using a Standard PCR Thermocycler with preheating for 5 min at 95 °C, followed by 35 cycles of denaturation for 1 min at 95 °C, annealing for 1 min at 60 °C, an extension for 1 min at 72 °C, and a final extension for 10 min at 72 °C. All surgery was performed under sodium pentobarbital anesthesia and all efforts were made to minimize suffering.

### 2.4. Determination of Zygosity by Real-Time Quantitative PCR (qPCR) of Genomic DNA

The transgene fragment positive mice were mated with the C57BL/6 mice. The F1 offspring with *Puf-A* shRNA heterozygous mice were then intercrossed with each other to generate homozygous, heterozygous, and non-transgenic F2 mice. All these transgenic mice were viable and fertile.

The zygosity of these F2 mice was determined with quantitative real-time PCR (qPCR). Genomic DNA was prepared from the tail fragments of each of the F2 mice. The qPCR reactions were conducted using the 7500 Fast Real-Time PCR System (Applied Biosystems). The qPCR entailed the following three-step cycling conditions: (i) 95 °C for 2 min, (ii) 40 cycles of 95 °C for 15 s, followed by 60 °C for 30 s, and (iii) 72 °C for 1 min. The reaction mixture (20 µL) consisted of 1 µL DNA solution containing 50–100 ng of genomic DNA, 1x SYBR Green Real-Time PCR Master Mix (Toyobo), and 6.25 pmol of each primer. The *Puf-A* shRNA transgene and a housekeeping *GAPDH* were amplified, respectively, as target and reference genes. The products of qPCR were tagged with SYBR Green. For the detection of the *Puf-A* shRNA transgene product, primer #3 (5′-TAG AGA ACG TAT GTC GAG TTT ATC C-3′) and primer #4 (5′-TGA AAG TAT TTC GAT TTC TTG GCT TTA TA-3′) were used. For the reference *GAPDH* gene, the forward primer (5′-GCC GTA TTG GGC GCC TGG TC -3′) and reverse primer (5′-GAC AAG CTT CCC ATT CTC GG-3′) were used.

The assessment of zygosity of these F2 transgenic mice had followed the previous report [26]. The cycle of threshold (Ct) values of *Puf-A* and *GAPDH* were obtained with qPCR of genomic DNA. First, the *Puf-A*/*GAPDH* ratio (ΔCt) was obtained by subtracting Ct of *GAPDH* from the Ct of *Puf-A* (ΔCt = Ct*_Puf-A_* − Ct*_GAPDH_*). Second, the ratio of *Puf-A*/*GAPDH* from F2 mice was normalized to F1 heterozygous founder mice by subtracting the *Puf-A*/*GAPDH* ratio from F1 and F2 mice (ΔΔCt = ΔCt_F2_ − ΔCt_F1_). Finally, the relative quantification of *Puf-A* shRNA transgene was given by the 2^–ΔΔCt^ method. Theoretically, the values of the 2^–ΔΔCt^ method obtained from heterozygous and homozygous F2 mice would be 1 and 2, respectively, but these ratios would deviate from the ideal values due to differences in the efficiency of the two primer sets used in qPCR [27].

### 2.5. Puf-A shRNA Induction

*Puf-A* shRNA is induced upon doxycycline (Dox) administration. The pregnant mice were treated with doxycycline (200 ng/mL) via intraperitoneal injection at E7.5 (embryonic day, the day after the vaginal plug was found) and dissected at indicated times. All surgery was performed under sodium pentobarbital anesthesia and all efforts were made to minimize suffering. The mice of *Puf-A* shRNA treated with and without doxycycline were referred to as the *Puf-A* knockdown and shRNA control, respectively.

### 2.6. Using Oct4-Driven EGFP Expression for Tracing PGCs

The *Oct4-EGFP* mice, which expressed an *Oct4* promoter-driven EGFP transgene (*GOF18∆PE-EGFP*), were obtained from The Jackson Laboratory, USA (stock no. 004654). All experimental procedures were performed in strict accordance with the Guide for the Care and Use of Laboratory Animals. The animal protocol was approved by the Institutional Animal Care and Use Committee of Chang Gung University. All surgery was performed under sodium pentobarbital anesthesia and all efforts were made to minimize suffering. The F2 homozygous *Puf-A* shRNA mice and *Oct4-EGFP^+/+^* mice crossed each other to generate the double mice with *Puf-A* shRNA^+/–^ *Oct4-EGFP^+/^*^–^.

### 2.7. Analysis of PGC Distribution in Embryos with Time-Lapse Video Microscopy

For the analysis of PGC distribution in whole-mount specimens, embryos were isolated at E9.0; furthermore, the head, the forelimb buds, and one flap of lateral epidermis were dissected to facilitate the visualization of PGCs within the hindgut. The dissected embryos were cultured in DMEM-F12 medium (Invitrogen, Waltham, MA, USA) supplemented with 15% FBS, penicillin/streptomycin (Invitrogen), non-essential amino acids (Invitrogen), L-glutamine (Invitrogen), and sodium pyruvate (Invitrogen). Then, the exposed trunk and tail of the embryo were cultured in a fresh culture medium with doxycycline (200 ng/mL). After 12 h, the whole assembly with E9.5 embryos was placed in a culture chamber at 37 °C on a confocal microscope stage. For time-lapse imaging, the caudal end of the embryo spanning the base of the allantois was embedded with the ventral side up in a 35 mm Petri dish with a thin layer of low-melt agarose to visualize the formation and migration of PGCs. To embed the dissected tissue in low-melt agarose, the Petri dish was first chilled on ice for 45–60 s; afterwards, 4–5 mL of pre-warmed (37 °C) medium was added. A small amount of mineral oil (Sigma) was used to cover the medium to prevent evaporation. The temperature was maintained at 37.0 ± 0.5 °C using a heated stage and time-lapse images were captured using fluorescence illumination under a Leica confocal microscope. Z-image stacks, comprising 10 μm slices spanning 200 μm thickness, were acquired every 10 min over 12 h.

### 2.8. Isolation of PGCs

The hindguts were explanted from E10.5 embryos. The gonads exhibiting *Oct4*-EGFP fluorescence were pooled and then dissociated into single cells by incubation with 0.05% Trypsin-EDTA in phosphate-buffered saline (PBS) at 37 °C for 10 min. When the cell suspensions were washed once in DMEM/F12 containing 10% FBS, and the cell clumps and tissue debris were removed by filtering, the cell suspensions were passed through a cell strainer (Falcon, Fisher Scientific, Thermofisher scientific, Taipei, Taiwan). Afterwards, the EGFP^+^ PGCs were directly sorted and collected into the PGC medium using fluorescence-activated cell sorting (FACS) by the AriaIII cell sorter (BD Biosciences, Stockholm, Sweden). The sorted PGCs were cultured in DMEM containing 10% FBS, 100 U/mL penicillin/streptomycin, and supplemented with growth factors (50 ng/mL soluble SCF from RD Systems Minneapolis; and 10 ng/mL LIF from Chemicon International). In the experiments in Figures 4 to 7, the PGCs were isolated by sorting *oct4*-EGFP^+^ cells from E10.5 embryos.

### 2.9. Cytospin and Immunofluorescence Staining

Samples were loaded into an automated cytospin machine (Shandon Cytospin 4, Thermo Electron Corporation, Thermofisher Scientific) following the manufacturer′s instructions, and centrifuged at 500–700 revolutions per min (rpm) for 5 min. The slides prepared by the cytospin technique were fixed by immersion in 95% ethyl alcohol for 20–30 min.

The primary antibodies for immunofluorescence staining were mouse anti-Puf-A [18], rabbit anti-p53 (sc-6243, Santa Cruz Biotechnology, Santa Cruz, CA, USA), and rabbit anti-NPM1 (sc-5564, Santa Cruz). The secondary antibodies for immunofluorescence staining were donkey anti-mouse IgG (Invitrogen) and Alexa555-conjugated donkey anti-mouse IgG (Jackson ImmunoResearch, West Grove, PA, USA). Briefly, the cells were fixed in 4% paraformaldehyde for 20 min at room temperature, permeabilized with 0.5% PBST for 5 min, and then blocked with 5% bovine serum albumin (BSA) for 30 min. Slides were incubated at 4 °C with primary antibodies. After overnight incubation, the cells were washed and incubated for 1 h at room temperature with secondary antibodies. The cells were then counterstained with DAPI (Sigma, Burlington, MA, USA).

### 2.10. BrdU Labeling

Pregnant female mice were given BrdU (50 mg/kg body weight at E10.5) by intraperitoneal injection, and E10.5 embryos were dissected from the pregnant female after 2 h of BrdU injection. The PGCs were isolated by sorting with *Oct4*-EGFP^+^ cells obtained from E10.5 embryos of the *Puf-A* knockdown and the control mice. Then, approximately 1 × 10^5^ isolated PGCs from mice were fixed in 4% paraformaldehyde for 15 min on ice and processed for immunofluorescence using the anti-BrdU antibody.

### 2.11. FACS Analysis of Cell Cycle

Single-cell suspensions containing *Oct4*-EGFP^+^ PGCs obtained from E10.5 embryos were prepared as described above and were fixed in 4% paraformaldehyde for 15 min on ice. Cells were incubated in 70% ethanol at −20 °C overnight and washed with FACS solution. Cells were then stained with 1 μg/mL DAPI for estimating the amount of DNA in the cells, and the cell cycle of EGFP^+^ PGCs was examined by flow cytometry. The average total number of PGCs analyzed by FACS was 1200. The cell cycle was analyzed from the DNA histograms using FlowJo software. The ratios of EGFP^+^ PGCs in the G1/G0, S, and G2/M phases were estimated using the Watson model.

### 2.12. Analysis of Cell Death

Approximately 1 × 10^5^ single-cell suspensions prepared from E10.5 embryos were fixed in 4% paraformaldehyde for 15 min on ice. The analysis of cell death was performed by TUNEL staining using the In Situ Cell Death Detection Kit, Fluorescein (Roche, Basel, Switzerland), according to the manufacturer’s procedure.

### 2.13. Flow Cytometric Analysis of Cell Apoptosis

Briefly, the PGCs were collected and suspended in a binding buffer at a density of 1 × 10^4^ cells/mL. Then, cells were prepared according to the instructions of the PE Annexin V Apoptosis Kit, and 100 μL of the solution was transferred to a 2 mL tube. Afterward, 5 μL of the phycoerythrin (PE) fluorescently labeled Annexin V and 5 μL of 7-amino-actinomycin D (7-AAD) were added and the tubes were incubated for 15 min at room temperature (25 °C) in the dark. Finally, we added 400 μL of 1X Binding Buffer to each tube (Roche). Flow cytometry was used to determine the percentage of Annexin V^+^/7-AAD^-^ cells and measure the cell surface expression of phosphatidylserine (PS) by Annexin V staining in early apoptotic PGCs. The PGC samples were analyzed for the presence of apoptotic cells by flow cytometry on a FACS flow cytometer (BD Biosciences). Apoptotic PGCs were identified as Annexin V^+^/7-AAD^-^ (early apoptotic) and Annexin V^+^/7-AAD^+^ (late apoptotic).

### 2.14. Sorting of PGCs and Quantitative RT-PCR

For qRT-PCR analysis, E10.5 embryos were dissected and incubated with 0.25% trypsin in PBS at 37 °C for 5 min, and then Dulbecco’s modified Eagle’s medium (DMEM) containing 10% fetal calf serum was added to stop the enzymatic reaction. After samples were pipetted, the singly suspended cells were collected by centrifugation (3000 rpm, 5 min) and resuspended in fluorescence-activated cell sorting (FACS) solution (2% fetal calf serum and 0.01% NaN3 in Hanks solution). EGFP^+^ cells were isolated using the AriaIII cell sorter (BD Biosciences).

Total RNA was extracted from the sorted cells using the RNeasy micro kit (Qiagen). Complementary DNA (cDNA) was synthesized using a PrimeScript RT reagent kit (Takara), and real-time PCR amplification was performed using a Thermal Cycler Dice (Takara) with SYBR Premix Ex Taq II (Takara). PCR conditions were 94 °C for 5 min, followed by 40 cycles at 94 °C for 15 s, and 60 °C for 30 s; the dissociation protocol was then conducted. The expression levels of mRNA were calculated and normalized to the levels of *GAPDH* mRNA. PCRs were performed using the following pairs of primers: *Puf-A*, 5′-GGTCA TGTGA GGAAG ATGCT GC-3′ and 5′-TTGTC CAGCG TTGGG TGATC TG -3′; *Lin28a*, 5′-GGTCT GGAAT CCATC CGTGT CA-3′ and 5′-TCCTT GGCAT GATGG TCTAG CC-3′; *pre-let7a*, 5′-TTCAC TGTGG GATGA GGTAG-3′ and 5′-TGGTT TCCTA TGAGA CCCCA T-3′; *Blimp1*, 5′-AAGAC GTTCG GTCAG CTCTC CA-3′ and 5′-CTGGC ACTCA TGTGG CTTCT CT-3′; *Oct4*, 5′-CAGCA GATCA CTCAC ATCGC CA -3′ and 5′- GCCTC ATACT CTTCT CGTTG GG-3′; *Sox2*, 5′- AACGG CAGCT ACAGC ATGAT GC -3′ and 5′- CGAGC TGGTC ATGGA GTTGT AC -3′; *Nanog*: 5′- GAACG CCTCA TCAAT GCCTG CA -3′ and 5′- GAATC AGGGC TGCCT TGAAG AG -3′; *Stella*, 5′- CCAAG AGAAG GGTCC GCACT TT -3′ and 5′- GCAGA GACAT CTGAA TGGCT CAC -3′; *Nanos3*, 5′- TCTGC AGGCA AAAAG CTGAC CC -3′ and 5′- GGGCT TCCTG CCACT TTTGG AA -3′; *p53*, 5′- TACCA CCATC CACTA CAAGT ACAT -3′ and 5′- CAGGG CAGGC ACAAA CACG -3′; *p21*, 5′- GACAA GAGGC CCAGT ACTTC -3′ and 5′- GCTTG GAGTG ATAGA AATCT GTC -3′; *Bax*, 5′- AGGAT GCGTC CACCA AGAAG CT -3′ and 5′- TCCGT GTCCA CGTCA GCAAT CA -3′; *PUMA*, 5′- ACCGC TCCAC CTGCC GTCAC -3′ and 5′- ACGGG CGACT CTAAG TGCTG C -3′; *GAPDH*, 5′-AAATG GTGAA GGTCG GTGTG-3′ and 5′- CATGT AGACC ATGTA GTTGA G -3′

### 2.15. Statistical Analyses

The results are expressed as mean ± SD. All experiments were repeated at least three times. Statistical evaluations were performed using Student′s *t*-test or a One-Way ANOVA test using GraphPad Prism version 5 (GraphPad Prism 5.00. https://www.graphpad.com/ (accessed on 24 April 2022)). A value of *p* < 0.05 was considered statistically significant.

## 3. Results

### 3.1. Generation of Transgenic Mice Carrying the Inducible Puf-A shRNA Construct

To examine the effect of *Puf-A* knockdown on PGC development and the maturation of germ cells, transgenic mice carrying the inducible *Puf-A* shRNA were generated. First, the annealed two oligonucleotides containing *Puf-A* shRNA were inserted into downstream of the modified Tet-responsive Pol III hybrid promoter (*P*_TREmod/U6_), which was cut at XhoI and HindIII restriction sites to construct the PGK-tTS-TRE-*Puf-A* shRNA vector (7.7 kb) (Figure 1A). This inducible knockdown system includes tTS protein that would bind to the tetO sequences within *P*_TREmod/U6_ to suppress the transcription of the *Puf-A* shRNA from the U6 promoter (Figure 1A) in the absence of doxycycline (Dox). Upon induction with doxycycline, the transcriptional suppression is relieved as tTS is dissociated from the *P*_TREmod/U6_, thus permitting the high-level transcription of *Puf-A* shRNA to repress *Puf-A* expression in mice.

The plasmid of PGK-tTS-TRE-*Puf-A* shRNA in Figure 1A was then excised with EcoRV and HindIII to produce the 3.4 kb transgenic fragment of *Puf-A* shRNA. This transgene fragment was introduced into the pronucleus of freshly fertilized oocytes of C57BL/6 mice. The microinjection of this transgene fragment yielded 21 newborn mice. To determine that if these mice carried *Puf-A* shRNA transgene, we performed PCR by a pair of specific primers (#1 and #2 primers in Figure 1A) using genomic DNA extracted from tail snips. Agarose gel electrophoresis revealed the presence of the 730 bp transgene fragments in six mice (Figure 1B).

### 3.2. Generation of the Inducible Puf-A shRNA and Oct4-EGFP Double Transgenic Mice

Afterwards, these six transgenic mice containing 730 bp in Figure 1B were mated with C57BL/6J mice individually to generate F1 heterozygous founder mice (*Puf-A* shRNA^+/–^) and non-transgenic mice (Figure 2A). The F1 heterozygous mice (*Puf-A* shRNA^+/–^) were regarded as founder mice with the germline integration of *Puf-A* shRNA transgene (left panel). These F1 founder mice mated with their sibling to generate F2 offspring in the genotypes of homozygous *Puf-A* shRNA, heterozygous *Puf-A* shRNA, and non-transgenic mice (right panel). All these transgenic mice were viable and fertile.

For zygosity determination of these F2 offspring (right panel in Figure 2A), we used a pair of transgene-specific primers (#3 and #4 primers in Figure 1A) to perform real-time quantitative PCR (qPCR) with the isolated genomic DNA extracts from F2 mouse tails. The threshold value (Ct) of the *Puf-A* shRNA transgene was normalized to *GAPDH*. The F1 heterozygous founder mouse was used as a calibrator. The results of qPCR analyses for these F2 mice with unknown zygosity revealed three clusters of qPCR data: (i) homozygote [*Puf-A* shRNA^+/+^; 2^-ΔΔCt^ value is 1.98 ± 0.23, *n* = 17] that scored around 2; (ii) heterozygote [*Puf-A* shRNA^+/–^; 2^-ΔΔCt^ value is 1.05 ± 0.26, *n* = 35] that scored around 1; and (iii) non-transgenic mice [*Puf-A* shRNA^–/–^; not detected, *n* = 15] (Figure 2B). Therefore, real-time qPCR data on genomic DNA showed three clusters of 2^-ΔΔCt^ for these three genotypes of F2 offspring.

To investigate the role of Puf-A in PGCs, we utilized the transgene, *Oct4-EGFP*, a marker for tracking PGCs in the mouse embryo [28]. The homozygous *Puf-A* shRNA mice (Figure 2A) were mated with *Oct4-EGFP^+/+^* mice, and all embryos from these pregnant mice were double heterozygous mice of *Puf-A* shRNA and *Oct4-EGFP* (*Puf-A* shRNA^+/–^*Oct4-EGFP^+/–^*) (Figure 2C). To reduce *Puf-A* levels, we used an intraperitoneal injection of doxycycline to express *Puf-A* shRNA at E7.5 (hereafter referred to as “*Puf-A* knockdown”). The PGCs from E10.5 embryos of the pregnant mice were isolated by FACS sorting for EGFP^+^ cells driven by *Oct4* promoter (Figure 2C). The result of qRT-PCR showed a decrease of approximately 80% in the expression of *Puf-A* mRNA in the PGCs from *Puf-A* knockdown embryos, as compared to the control PGCs isolated from the embryos of *Oct4-EGFP^+/+^* treated with doxycycline (hereafter referred as “Dox control”) or the embryos of *Puf-A* shRNA^+/–^*Oct4-EGFP^+/–^* in the absence of doxycycline administration (hereafter referred as “shRNA control”) (Figure 2D). These results indicate the efficient reduction of *Puf-A* expression in PGCs from the embryos of *Puf-A* shRNA^+/–^*Oct4-EGFP^+/–^* upon doxycycline treatment.

### 3.3. Puf-A Is Required for PGC Growth during Gonad Development

To investigate if Puf-A participates in PGC development, we observed the EGFP-expressing PGCs driven by *Oct4* promoter at indicated times of embryonic development (Figure 3A). In Dox control embryos, the EGFP+ PGCs were displayed in the caudal region at E8.5, and proliferation of these cells spanned to the hindgut region at E9.5 (Figure 3B). At about E10.5, the PGCs proliferated further and rapidly (Figure 3B). At the same time, the PGCs started to leave the hindgut and migrate toward bilateral genital ridges. Colonized PGCs within the genital ridges developed in male and female gonads (Figure 3B). The PGC growth in the embryos of shRNA control was indistinguishable from that of the Dox control embryos (Figure 3C).

*Puf-A* shRNA was overexpressed with Dox administration via intraperitoneal injection to E7.5 pregnant mice. At E9.5, there was only a slight reduction in the number of PGCs in *Puf-A* knockdown embryos (Figure 3D). However, at E10.5, very few PGCs were found in the gut tube and genital ridges in *Puf-A* knockdown embryos (Figure 3D). These results reveal a significant loss of PGCs during embryonic development upon *Puf-A* knockdown.

The migration of PGCs over time is a critical step for gonad development. We used time-lapse images of PGC migration in embryos during E9.5 to E10.5 to observe the influence of *Puf-A* knockdown (see Appendix A). The *Oct4*-EGFP-expressing PGCs were actively motile throughout the hindgut in both control embryos (Appendix A). Moreover, most PGCs became motionless in the *Puf-A* knockdown embryos (Appendix A), indicating that Puf-A is required for PGC movement.

### 3.4. NPM1 Translocation to Nucleoplasm upon Puf-A Knockdown in PGCs

A previous study in cancer cells has reported that Puf-A is a nucleolar protein and is associated with the nucleolar protein, nucleophosmin (NPM1) [18]. In this study, the subcellular localization of Puf-A in PGCs was examined with immunostaining of the isolated PGCs from E10.5 embryos using anti-Puf-A antibody. As shown in Figure 4A,B, the PGCs obtained from embryos of Dox and shRNA controls showed that Puf-A (green) was distributed in the nuclei in a manner consistent with the nucleolus location. More importantly, the immunofluorescent staining also showed that NPM1 was also found in the nucleoli (Figure 4A,B). In contrast, upon *Puf-A* knockdown, the green staining using anti-Puf-A antibody was not observed in PGCs, confirming the specificity of this antibody and *Puf-A* knockdown efficiency (Figure 4C). Furthermore, NPM1 was distributed throughout the nucleoplasm of the *Puf-A* knockdown PGCs (Figure 4C). Therefore, these results agreed with the previous observation of NPM1 translocation from the nucleolus to nucleoplasm in the cancer cells in *Puf-A* knockdown cells [18].

### 3.5. Knockdown of Puf-A Activates p53 Signaling in PGCs

The nucleoplasmic localization of NPM1 from the nucleolus was reminiscent of the stabilization of p53 through competitive interaction with MDM2 [29]. Our result showed a remarkable increase in p53 proteins in the PGCs from embryos of *Puf-A* knockdown, while p53 remained undetectable in the PGCs from the other two control embryos (Figure 5A). However, the levels of *p53* mRNA from the PGCs of the *Puf-A* knockdown, Dox control, and shRNA control embryos were not significantly different (Figure 5B). Thus, these results suggest that silencing of *Puf-A* expression is accompanied by an accumulation of p53 through post-transcriptional regulation by the stability of p53 protein.

p53 is a major tumor suppressor that prevents tumor formation in damaged or stressed cells by promoting the transcription of target genes, *p21*, *PUMA*, and *Bax*, which are associated with cell cycle arrest and apoptosis [30,31]. To corroborate that PGCs harbor dysregulated p53 signaling upon *Puf-A* knockdown, we measured p53 target genes, *p21*, *Bax*, and *PUMA* by performing qRT-PCR from the isolated PGCs at E10.5. The result showed a significant increase in the expression of these genes upon *Puf-A* knockdown (Figure 5B), consistent with the suggestion of strong activation of p53 signaling in PGCs when *Puf-A* was knockdown.

### 3.6. Puf-A Knockdown Impairs the Growth of PGCs

Rapid amplification in cell number of PGCs is crucial for gonad development. Since the knockdown of *Puf-A* led to large PGC loss during gonad development and up-regulation of *p21* following the p53 activation, *Puf-A* knockdown might induce cell cycle arrest of PGCs. In Figure 6A, we analyzed the cell proliferation status of PGCs isolated from E10.5 embryos using the BrdU incorporation assay. As shown in Figure 6A, the proportion of BrdU-labeled cells in PGCs was markedly reduced upon *Puf-A* knockdown, when compared with the PGCs from the Dox control and the shRNA control.

To further examine the cell cycle status of DNA in PGCs isolated from E10.5 embryos, we used DAPI staining to perform an FACS analysis. The percentages of PGCs, in G1/G0, S, and G2/M phases were quantified with FlowJo software. The results showed that there was a two-fold increase in the proportion of G1 phase in the PGCs obtained from *Puf-A* knockdown mice (73.1%) compared with the control mice (about 37% in the controls). Conversely, the percentage of PGCs at S phase was markedly reduced from 46.9 and 50.8% in both controls to 13.07% (Figure 6B). On the other hand, there was no significant difference in the proportion of cells at the G2 phase for *Puf-A* knockdown PGCs (13.2%) and the control PGCs (13−16%). These results suggest the requirement of Puf-A in PGC proliferation, especially the G1/S transition during cell cycle progression.

The expressions of the *Bax* and *PUMA* genes are regulated by p53 and involved in p53-mediated apoptosis [30,31]. Thus, we examined apoptosis in PGCs isolated from E10.5 embryos following *Puf-A* knockdown by performing TUNEL assays. As shown in Figure 6C, PGCs were examined with TUNEL staining (red). PGCs isolated from *Puf-A* knockdown embryos displayed more TUNEL-positive cells (red in Figure 6C). Furthermore, the results of flow cytometric analysis with the positive of Annexin V-PE showed a three-fold increase in PGCs upon *Puf-A* knockdown (64.5%) compared with the controls (about 20%) (Figure 6D), consistent with the suggestion that Puf-A has anti-apoptotic activity.

To further corroborate the requirement of Puf-A for PGC growth, we also examined the growth of PGCs in vitro (Figure 6E). The PGCs were isolated from E10.5 embryos and seeded at a density of 10,000 cells per well in a 12-well plate. The isolated PGCs from these embryos were cultured for 72 h in the presence of 0.2 μg/mL doxycycline (Dox control and *Puf-A* knockdown) or its absence (shRNA control). The growth curves of PGCs using trypan blue dye exclusion revealed that these two control PGCs proliferated continuously over time, but the growth of *Puf-A* knockdown PGCs declined at 72 h (Figure 6E). It is consistent with the results that Puf-A is critical for cell cycle progression and the survival of PGCs.

### 3.7. Puf-A Protects Germ Cell Fate during Early Gonad Development

PGCs are specialized around E7.5, but continuously express the markers associated with pluripotent genes and germ cell specification factors until E12 [32]. To investigate whether knockdown of *Puf-A* affects the germ cell fate, we examined the expression of these pluripotency and germline specification markers in PGCs isolated from E10.5 embryos. First, PGCs from the *Puf-A* knockdown embryos expressed significantly lower pluripotent markers, such as *Oct4*, *Sox2*, and *Nanog,* than the control embryos (Figure 7A). The mRNA levels of the regulators of germline specification, such as *Lin28* and *Blimp1,* were down-regulated, but *pre-let-7* miRNA was up-regulated, which became *let-7* microRNA to inhibit germline differentiation (Figure 7B). Furthermore, genes associated with germ cell development, such as *Stella* and *Nanos-3*, were reduced upon *Puf-A* knockdown (Figure 7C). These results are consistent with previous studies that Lin28 promotes Blimp1 expression by inhibiting *let-7* microRNA maturation, and Blimp1 is a master regulator for PGC specification and germline development by increasing the levels of these pluripotent markers and genes involved in germ cell development [21,33,34,35,36,37]. Thus, these results suggest the importance of Puf-A in the maintenance of germ cell fate for PGCs during gonad development.

## 4. Discussion

Previously, the zebrafish (*Danio rerio*) was used as a model to study the functions of 13,711 new Ka/Ks-predicted human exons, which were identified by comparative evolutionary genomics analysis [16]. Puf-A was predicted to be a new protein-coding exon and was characterized in zebrafish, mice, and humans [16]. Recently, we also reported that Puf-A is a novel assembly factor involved in ribosome biogenesis and is required for cancer growth and tumor progression through its interactions with NPM1 [18]. In the present study, we have described the biologic functions of Puf-A for PGCs during gonad development. First, Puf-A is required for the rapid amplification and active migration of PGCs for gonad development. Second, the nucleoplasmic translocation of NPM1 from nucleolus and the dysregulated p53 signaling might contribute to the loss of PGCs upon *Puf-A* knockdown. Finally, the expression of Puf-A is important for germ cell identity in gonad development to ensure proper PGC growth and differentiation.

In the mouse embryo, PGCs are detectable at E7.5 and then become motile, starting to migrate from the hindgut (E9.0), along the dorsal mesentery (E9.5), and finally reaching the lateral genital ridge [38]. Here, we showed that the movement of the PGCs in *Puf-A* knockdown embryos was significantly slowed down during migration, implying that these motionless PGCs would have difficulty moving to the genital ridge. At these stages of embryo development, the PGCs displayed pluripotent markers and express germline specification factors; however, the germ cell fate of these PGCs could be unstable and the lineage could be lost after further specification [39]. In current studies, we showed that *Puf-A* knockdown resulted in the down-regulation of *Lin28* and *Blimp1* and the up-regulation of *Pre-let-7* miRNA. This agrees with previous studies in that the transcription factor Blimp1 is the *Pre-let-7* target, a master regulator of PGC specification from E7 to E13.5, and its down-regulation leads to the impairment of proliferation and migration, thus resulting in a significant loss of PGC cells [21,33]. Furthermore, Lin28 is the regulator required for PGC formation via the suppression of *Pre-let-7* microRNA for the Blimp1-mediated development of germ cell lineage [34]. Therefore, a decrease in Lin28 would markedly reduce the generation of PGCs [33]. In *Blimp1* knockout embryos, PGC specification remained at the early stage of development, displaying low levels of Stella and Nanos3 [21]. In addition, the conditional knockout of *Oct4* or *Nanog* in the migrating PGCs were reported to result in the death of PGCs by apoptosis, indicating the importance of these genes for PGC maintenance [35,36]. In addition, embryos lacking Sox2 in the PGCs showed a dramatic decrease in germ cell numbers, due to a reduction in cell proliferation [37]. Thus, it is conceivable that Puf-A is required for the maintenance of PGCs via the regulation of these essential genes involved in embryo development. Previous studies have shown that colonization of gonads by PGCs is a prerequisite for PGC differentiation [40]. In the E13.5 embryos of the control mice, the gonadal ridges filled with PGCs, while only few PGCs found in the nascent gonads in *Puf-A* knockdown embryos, suggesting that PGCs failed to colonize the gonads upon silencing *Puf-A* expression. Thus, it is likely that PGC differentiation was blocked, and the development of gonads was dysfunctional.

Several studies have demonstrated the interaction between Puf-A and 5.8S rRNA [16,17]. NPM1 in nucleoplasm promotes the accumulation of p53 proteins by disrupting the interaction between p53 and MDM2 [29,41]. Furthermore, we previously showed that in cancer cells, Puf-A interacted with NPM1 in nucleolus, based on studies of immunofluorescence staining and immunoprecipitation [18]. Current studies further demonstrate the nucleoplasmic translocation of NPM1 in *Puf-A* knockdown PGCs. Since NPM1 could interact with other nucleolar proteins, it will be of interest to pursue whether the interaction between other nucleolar proteins and NPM1 would be affected by *Puf-A* knockdown, but this latter issue will need further investigation. It is likely that mechanistically, *Puf-A* silencing would alter NPM1 localization, leading to the changes of the retention of ribosome proteins in nucleolus and diminish ribosome biogenesis, followed by cell cycle arrest/cell death. On the other hand, the nucleoplasmic translocation of NPM1 s might explain the accumulation of p53 proteins in *Puf-A* knockdown PGCs, subsequently causing the activation of downstream targets, including *p21*, *Bax*, and *PUMA*. Finally, all of these events lead to an increase in cell cycle arrest at G1 and apoptosis of PGCs. Overall, our findings showed Puf-A plays a critical role in the gonad development of PGC growth.

## 5. Conclusions

Here, we generated transgenic mice carrying inducible *Puf-A* shRNA and obtained the double heterozygous mice of *Puf-A shRNA* and *Oct4-EGFP* to examine the behavior of PGCs embryo development in mice. Our results showed that Puf-A played a protective role for PGC development by regulating their proliferation, survival, migration, and differentiation. We also found the alteration of NPM1 location and p53 expression upon reducing *Puf-A* level to be consistent with a previous study in cancer cells, which might explain the molecular mechanism of Puf-A functions in PGC development in mice.

## Figures and Tables

**Figure 1 cells-11-01476-f001:**
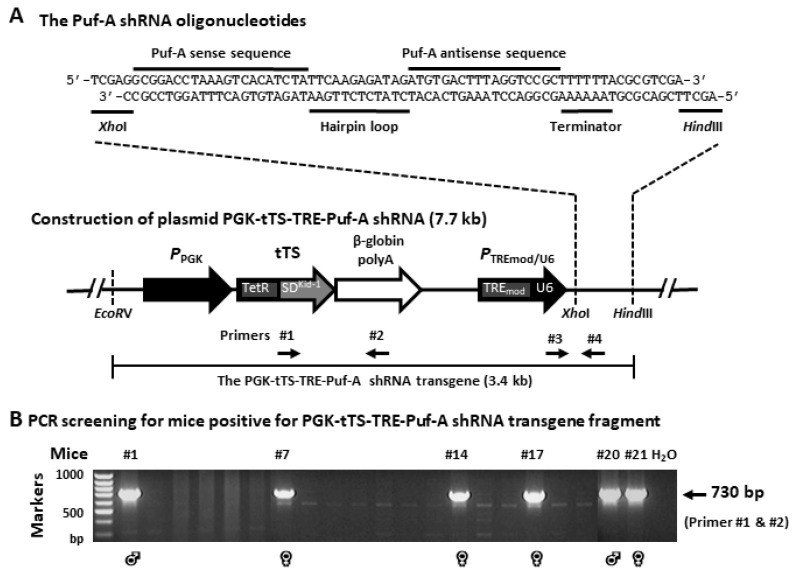
Generation of transgenic mice carrying the inducible *Puf-A* shRNA construct. (**A**) Schematic presentation of the construct used to generate inducible *Puf-A* shRNA transgenic mice. The complementary *Puf-A* shRNA oligonucleotides were inserted into the plasmid PGK-tTS-TRE. The fragment of the *Puf-A* shRNA transgene was in the region between EcoRV and HindIII. Primers #1 and #2 were used for detection in C57BL/6J mice that were positive for the inducible *Puf-A* shRNA transgene (730 bp). Primers #3 and #4 were used for real-time qPCR from genomic DNA of the transgenic mice to determine the zygosity of the mice with *Puf-A* shRNA. (**B**): Genotyping of the 21 mice injected with *Puf-A* shRNA transgene used primers #1 and #2 (Figure 1A) in the PCR reaction. Amplification of *Puf-A* shRNA transgene (arrows, 730 bp) was detected in mice #1 (♂), 7 (♀), 14 (♀), 17 (♀), 20 (♂) and 21 (♀). Distilled water was used as the negative control.

**Figure 2 cells-11-01476-f002:**
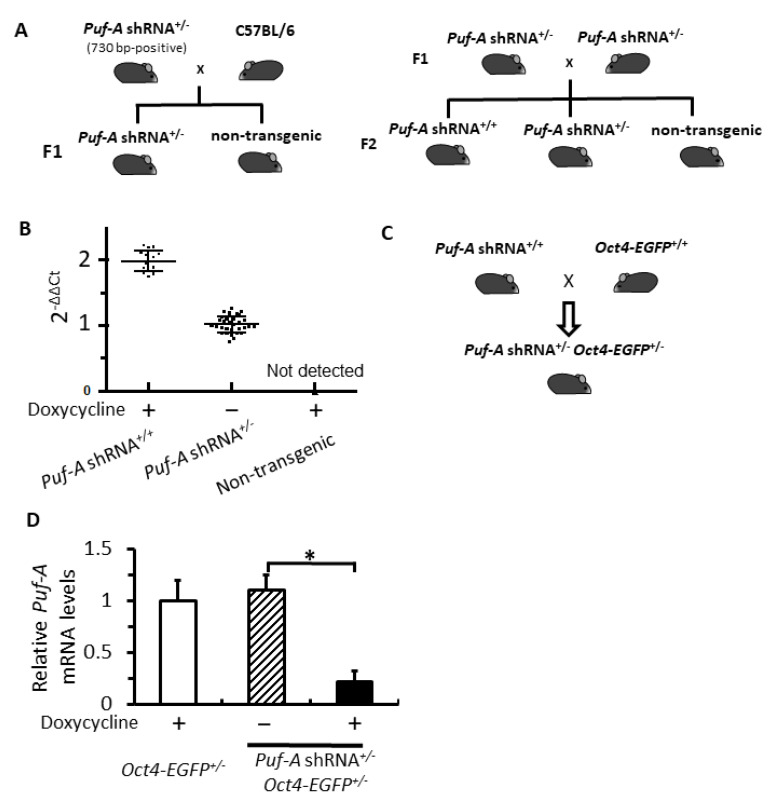
Generation of the inducible *Puf-A* shRNA and *Oct4-EGFP* double transgenic mice. (**A**) Breeding scheme of the *Puf-A* shRNA homozygous transgenic mice. (**B**) The real-time qPCR reactions for *Puf-A* shRNA transgene using primers #3 and #4 in Figure 1A were performed using genomic DNA of the F2 mice and normalized to *GAPDH* gene. Each dot represented the qPCR result from one mouse. The data are shown as mean ± SD. (**C**) Breeding scheme to generate the double transgenic mice of *Puf-A* shRNA^+/^^–^ *Oct4-EGFP^+/–^*. (**D**) qRT-PCR analysis for *Puf-A* in PGCs from E10.5 embryos. The PGCs from *Puf-A*-silencing embryos exhibited approximately 80% knockdown of *Puf-A* mRNA expression. The data are presented as mean ± SD (* *p* < 0.05, *n* = 3).

**Figure 3 cells-11-01476-f003:**
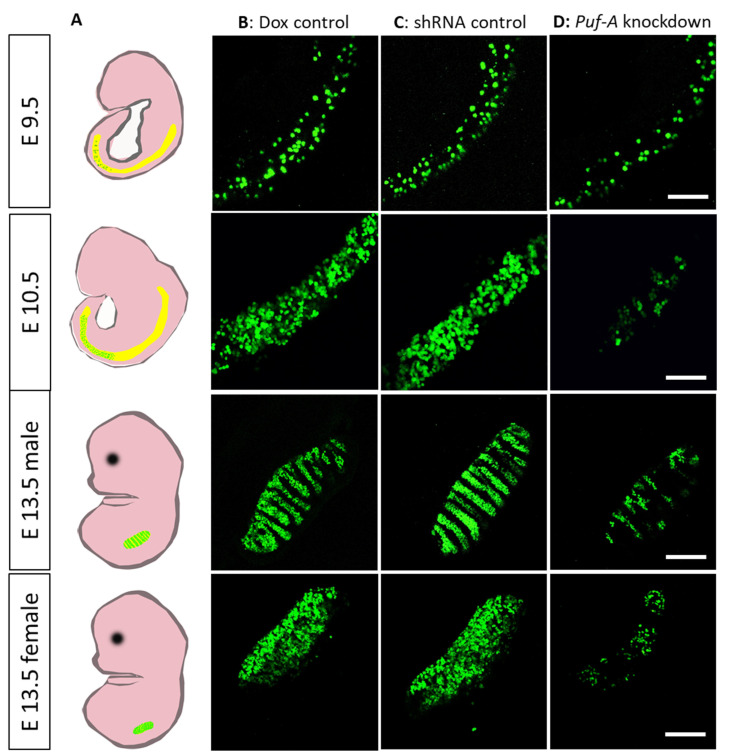
Puf-A is required for PGC growth during gonad development. (**A**) Schematic presentation for the location of PGCs in mouse embryos at E9.5, E10.5, and E13.5. PGCs were found in gut tube at E9.5; then they proliferated at E10.5. At E13.5, these PGCs migrated to lateral genital ridges and colonized in gonads. Green dots indicated PGCs. (**B**,**C**) The *Oct4*-EGFP^+^ PGCs from the embryos of Dox control (**B**) and shRNA control (**C**) mice showed plentiful cell growth in the hindgut tube of E9.5 to E10.5 embryos, and finally colonization of PGCs in genital ridges of male and female embryos at E13.5. (**D**) The *Oct4*-EGFP^+^ PGCs from the *Puf-A* knockdown embryos at E9.5 showed a slight decrease in cell number in the hindgut; in contrast, at E10.5 and E13.5, very few PGCs were observed. At least 12 embryos were used for each time points and with each genotype. Scale bars: 100 μm.

**Figure 4 cells-11-01476-f004:**
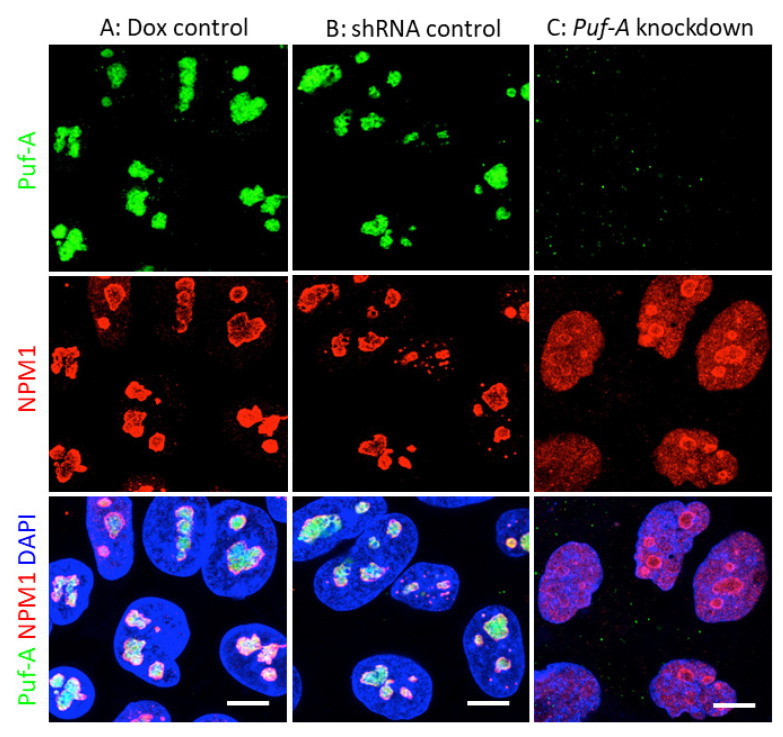
Puf-A is localized in the nucleolus of PGCs. PGCs were immunostained with anti-Puf-A antibody (green), then co-stained with anti-NPM1 antibody (red) and DAPI (blue). (**A**,**B**) PGCs from the Dox control (**A**) and shRNA control (**B**) embryos showed that Puf-A and NPM1 were co-localized and formed multiple clusters within the nucleus. (**C**) PGCs from embryos of *Puf-A* knockdown mice had no Puf-A expression and NPM1 was dispersed throughout the nucleoplasm. Scale bars, 10 μm.

**Figure 5 cells-11-01476-f005:**
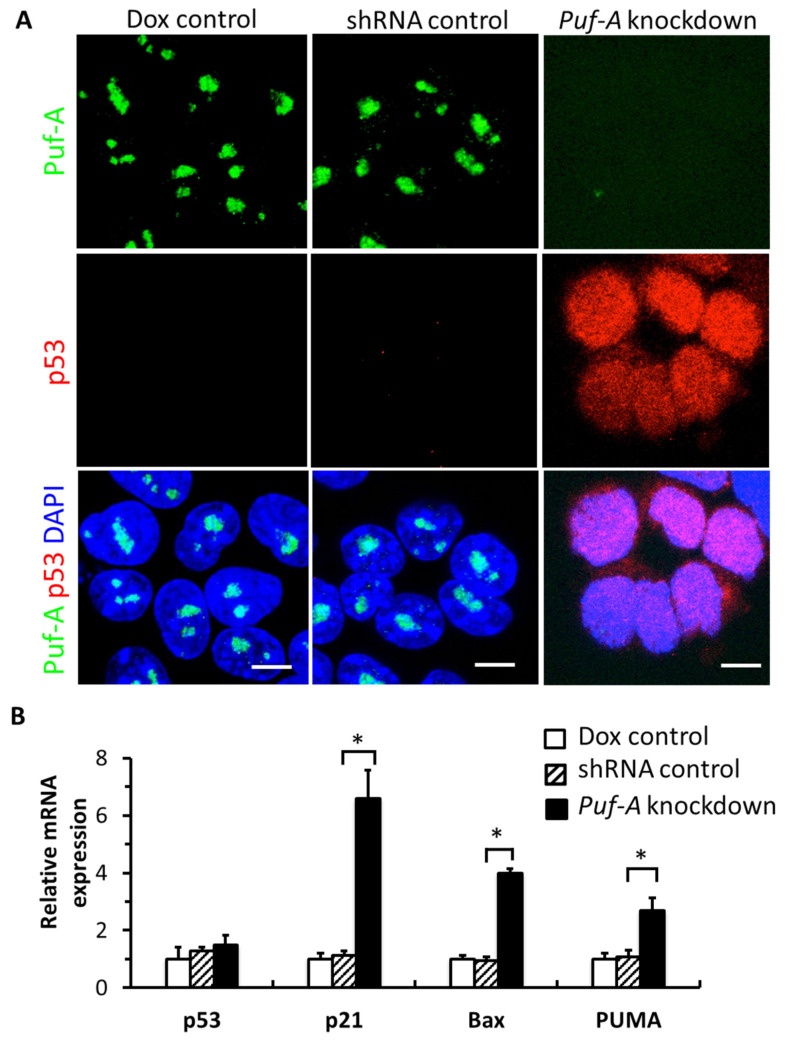
*Puf-A* knockdown activates p53 signaling in PGCs. (**A**) PGCs were immunostained with anti-p53 antibody (red), co-stained with anti-Puf-A antibody (green) and DAPI (blue). Extremely low levels of p53 signals were observed in PGCs from both embryos of Dox control and shRNA control; however, Puf-A aggregates were observed in the nucleus. PGCs from embryos of *Puf-A* knockdown had no Puf-A expression but showed high level of p53 in the nucleus. Scale bars, 10 μm. (**B**) qRT-PCR analyses for *p53* and p53 downstream targets, *p21*, *Bax*, and *PUMA* in the PGCs from E10.5 embryos. The levels of *p53* mRNA were unchanged in PGCs from *Puf-A* knockdown embryos, but the expression of *p21*, *Pax*, and *PUMA* mRNA levels was significantly increased. *GAPDH* was the internal control, and each gene was further normalized to Dox control PGCs. Bars are displayed as mean ± SD (* *p* < 0.05, *n* = 3).

**Figure 6 cells-11-01476-f006:**
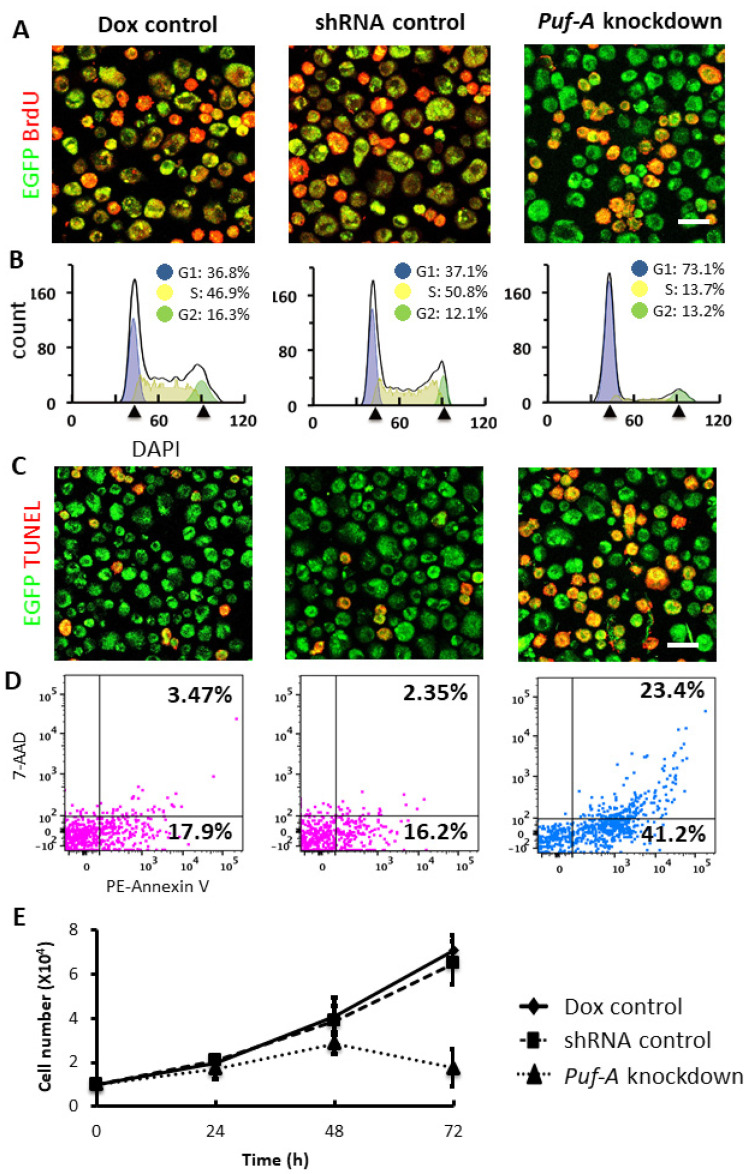
*Puf-A* knockdown impairs the growth of PGCs. (**A**): BrdU incorporation was conducted to detect the proliferation of PGCs. Most *Oct4*-EGFP^+^ PGCs (green) from Dox control and shRNA control embryos, were labeled with BrdU (red); however, upon *Puf-A* knockdown, less than a half of PGCs had BrdU incorporation. Scale bars, 20 μm. (**B**) Flow cytometry analysis of the cell cycle distribution of PGCs. The percentage of PGCs in G1 phase increased from 37% (controls) to 73.1% (*Puf-A* knockdown). The percentage of PGCs in S phase decreased from 48.9% (controls) to 13.7% (*Puf-A* knockdown). (**C**) TUNEL labeling was performed to measure the apoptosis of PGCs. Half of *Oct4*-EGFP^+^ PGCs (green) from *Puf-A* knockdown embryos were labeled with TUNEL (red), while very few PGCs had TUNEL labeling in Dox control and shRNA control embryos. Scale bars, 20 μm. (**D**) Cell apoptosis and death analysis using PE Annexin V/7-AAD kit. The percentage of apoptosis of PGCs (Annexin V^+^) increased from 20% (controls) to 64.5% (*Puf-A* knockdown). (**E**) In vitro cell growth assay of the PGCs. The PGCs from both Dox control and shRNA control embryos continued to grow and the number of PGCs increased more than six-fold after 72 h of in vitro culture. In contrast, the number of *Puf-A* knockdown PGCs started to decline after 48 h of culture.

**Figure 7 cells-11-01476-f007:**
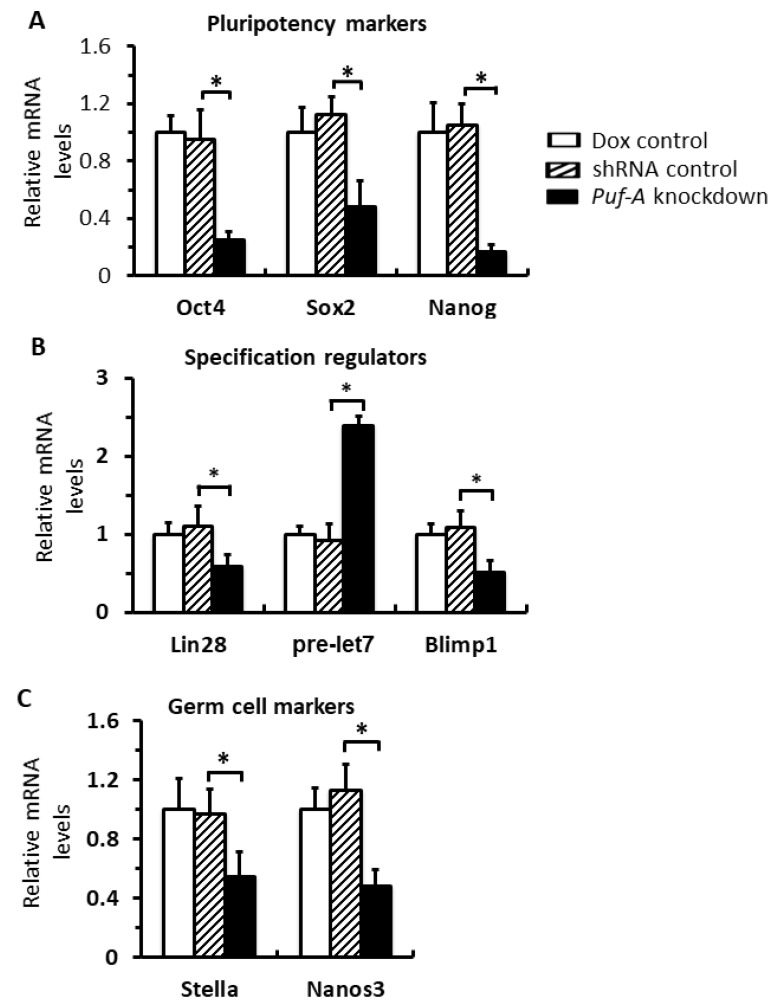
Puf-A protects germ cell fate during early gonad development. qRT-PCR analysis for the PGCs from E10.5 embryos. In *Puf-A* knockdown PGCs, the expression of genes associated with (**A**) pluripotency markers (*Oct4*, *Sox2*, and *Nanog*), (**B**) specification regulators (*Lin28* and *Blimp1*), and (**C**) germ cell markers (*Stella* and *Nanos3*) was significantly decreased. However, *pre-let-7*, the let-7 precursor, which is repressed by Lin28, was up-regulated upon *Puf-A* knockdown. GAPDH was used as internal control, and Dox control was as also the control. Error bars represent SD with *n* = 3, and * *p* < 0.05.

## Data Availability

Not applicable.

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
