# Peer review of "The Puf-A Protein Is Required for Primordial Germ Cell Development"

_cells, 2022, doi:10.3390/cells11091476_

Round 1

Reviewer 1 Report

The research article entitled ‘The Puf-A protein is required for primordial germ cell 2 development’ by Ko et al., attempts to characterize role of Puf-A protein in development of primordial germ cell (PGC) in mammalian cells. They have developed transgenic mice carrying inducible Puf-A shRNA and obtained the double heterozygous of Puf-A shRNA and Oct4-EGFP to examine the behavior of PGCs. They showed knockdown of Puf-A led to a loss of large number of PGCs, and the movement of the remaining PGCs slowed down. They have also reported nucleoplasmic translocation of NPM1 from nucleolus and hyperactivation of p53 in PGCs, when Puf-A was silenced. They have shown PGCs with decreased expression of essential genes associated in maintenance of PGC, increased subpopulation of G1 arrest and apoptosis in Puf-A knockdown embryos. Overall, their study showed that Puf-A governed PGC development by regulating PGC growth, survival, and maintenance. This manuscript will be of interest to the general readers of Cells and would be of great importance to the people in Cancer research and therapeutics field. However, I recommend this article for publication after some modifications which are written below.

Minor Comments:

  1. The figure legends for all figures are very lengthy and redundant with the results section. The legends should be more concise.
  2. Figure 4: How the authors could explain the nucleoplasmic translocation of NPM1 in Puf-A knockdown cells where NPM1 could interact with other nucleolarproteins? The authors have not shown also if the nucleolar structure remain intact in Puf-A knockdown cells. The loss of nucleolar integrity may also result distributed stain of NPM1. Also they have explained the interaction between NPM1 and Puf-A as they have obtained in cancer cells in previous study. How they can corelated this fact in context to the present study?
  3. The authors should comment on differential nucleolar structure as observed by Puf-A staining for Figure 5 between dox control and Sh RNA control. Also, they should explain how p53 expression could be differed in these two cell lines.
  4. The authors have shown Puf-A is important for PGC growth and development, but if the defects associated with the knock down impacts the development of gonad or the formation of mature germ cells, is not clear.

Reviewer 2 Report

In this MS, Ko et al. describe the role of Puf-A in the development of primordial germ cells (PGCs) in mice and suggest its involvement in mice gonad development. In particular, they show that Puf-A appears involved in mice PGC growth, survival and maintenance, and their inducible Puf-A knockdowns thus help hypothesize the molecular mechanisms for the role of Puf-A in mice PGC development. They show that a loss of Puf-A results in problems for PGCs to cross the G1/S transition during cell cycle progression, leads to the abnormal translocation of a nucleolar protein NPM1 and the aberrant activation of p53, a molecule with implications in aberrant cell division and thus cancer. Overall, this is a good contribution to the field and the MS can be accepted following minor revisions.

Comments:

  1. For fig3 (line 369-377) in MS, authors have forgotten to mention fig. subparts/subpanels both in description and well as in figure. Please correct that.

Fig.3: Can the authors provide quantifiable statistical data to show the lack of PGC differences between shRNA control and Dox control PGCs? (i.e. quantifiable data other than microscopy for sub panels across fig3?). Also why do the male E13.5 day embryos have such a striated appearance compared to same aged female embryos? Also can the authors provide a schematic of mouse embryos showing the PGCs and locations being discussed. The authors can additionally use arrows or pointers of some sort in the different sub panels to draw the eye of the reader 

  1. Specificity of anti-puf-A antibody used - evidence it is not binding to other PUM domain proteins within mice and/or interspecies specificity use
  2. Please label which sub panels the authors are referring to also for Fig4 (e.g. results section line 407). Also can the labelling (Puf-A/DAPI/NPM1) be moved outside the image box so that it doesn’t obstruct the image?
  3. Any ISH data to show levels of p53 mRNA remaining constant in control vs Puf-A knockdown conditions?
  4. Why were p21, Pax and PUMA selected as genes (over others) to be tested along side p53? Can the authors include the rational in the figure legends as well
  5. Any sequencing data to validate molecular pathways and DEG discussed in the MS?
  6. In fig6A (Puf-A knockdown panel, top right): why do the number of PGCs look comparable in this panel vs control if Puf-A knockdowns decrease PGC number? Can the figure be labelled for embryonic stage as well?

Round 2

Reviewer 1 Report

The manuscript entitled ‘The Puf-A protein is required for primordial germ cell development' can be accepted with some minor revisions which are stated below. This article presents interesting data for role of a RNA binding protein Puf-A in regulating growth and maintenance of development of Primordial Germ Cells (PGC). Earlier studies have shown function of Puf-A in controlling germ cells fate in Puf-A knockdown cells in Zebra fish and being a conserved protein function of Puf-A is believed to be conserved. This study has shown the function of Puf-A in the formation of PGC by generating several transgenic mice carrying inducible Puf-A shRNA. The authors have stratified most of the queries raised earlier. However, I still feel the authors should try to avoid repetitive sentence from figure legends. I have some minor comments which are stated below. The article can be accepted for publication after the authors resolve these minor comments.

  1. ‘The complex of ASH1 mRNA and Puf6 is translocated to the distal tip of the growing daughter cell, suggesting that yeast Puf6 translocate in the cytoplasm when yeast budding’- should be reframed. The formation of sentence is not right.
  2. The legends of the figure should describe the figure only, not the method. The authors are requested to shorten the legend for figure 6. Also please avoid the statements which should be in the method section. For example,” The PGCs were isolated from the E10.5 embryos by sorting with Oct4-EGFP+ cells.” in Figure 6 or in Figure 4 or 5.

Author Response

Response to Reviewer 1 Comments

Point 1: The complex of ASH1 mRNA and Puf6 is translocated to the distal tip of the growing daughter cell, suggesting that yeast Puf6 translocate in the cytoplasm when yeast budding’- should be reframed. The formation of sentence is not right.

Response 1: We thank the reviewer for pointing out this error. We have revised the sentence (lines 45-49) as follows:

“In budding yeast, Puf6 also functions as a translation repressor of ASH1 via directly binding to its mRNA 3’UTR, and the effect of translation repression by Puf6 is required for ASH1 mRNA asymmetric segregation during anaphase of cell cycle.”

Point 2: The legends of the figure should describe the figure only, not the method. The authors are requested to shorten the legend for figure 6. Also please avoid the statements which should be in the method section. For example,” The PGCs were isolated from the E10.5 embryos by sorting with Oct4-EGFP+ cells.” in Figure 6 or in Figure 4 or 5.

Response 2: We appreciate the reviewers for these suggestions to make this manuscript better. We have deleted the sentence “The PGCs were isolated from the E10.5 embryos by sorting with Oct4-EGFP+ cells.”. In addition, we have rewritten the legends of Figure 6 in the revised manuscript (lines 496-533). Furthermore, we also shortened the legends of Fig. 1-4 (lines 298-324, lines 342-368, lines 404-421, and lines 448-458).